# Study of Human Thermal Comfort for Cyber–Physical Human Centric System in Smart Homes

**DOI:** 10.3390/s20020372

**Published:** 2020-01-09

**Authors:** Yuan Fang, Yuto Lim, Sian En Ooi, Chenmian Zhou, Yasuo Tan

**Affiliations:** 1Japan Advanced Institute of Science and Technology (JAIST), 1-1 Asahidai, Nomi, Ishikawa 923-1292, Japan; sianen.ooi@jaist.ac.jp (S.E.O.); zhou_cm@jaist.ac.jp (C.Z.); ytan@jaist.ac.jp (Y.T.); 2School of Information Science and Engineering, Dalian Polytechnic University (DPU), No.1 Qinggongyuan, Dalian 116034, China

**Keywords:** human thermal comfort, heart rate, smart homes, cyber–physical systems

## Abstract

An environmental thermal comfort model has previously been quantified based on the predicted mean vote (PMV) and the physical sensors parameters, such as temperature, relative humidity, and air speed in the indoor environment. However, first, the relationship between environmental factors and physiology parameters of the model is not well investigated in the smart home domain. Second, the model that is not mainly for an individual human model leads to the failure of the thermal comfort system to fulfill the human’s comfort preference. In this paper, a cyber–physical human centric system (CPHCS) framework is proposed to take advantage of individual human thermal comfort to improve the human’s thermal comfort level while optimizing the energy consumption at the same time. Besides that, the physiology parameter from the heart rate is well-studied, and its correlation with the environmental factors, i.e., PMV, air speed, temperature, and relative humidity are deeply investigated to reveal the human thermal comfort level of the existing energy efficient thermal comfort control (EETCC) system in the smart home environment. Experimental results reveal that there is a tight correlation between the environmental factors and the physiology parameter (i.e., heart rate) in the aspect of system operational and human perception. Furthermore, this paper also concludes that the current EETCC system is unable to provide the precise need for thermal comfort to the human’s preference.

## 1. Introduction

A home is the most crucial place that provides a safe living environment and comfort for the human to meet people’s physical and psychological needs. It is a place not only where people gather with families and friends, but also people can relax, do any activities, obtain entertainment and enjoyment, and go to sleep. Smart homes that represent a branch of ubiquitous computing involves incorporating smartness into houses for comfort, healthcare, safety, security, and energy conservation. cyber–physical home system (CPHS) comprises many smart home systems for a variety of services and applications that applies the concept of cyber–physical systems (CPSs) [1] in the smart home domains to provide home automation control, especially aiming for comfortability, such as healthcare monitoring, energy savings. Today, active researches on CPHS have led to a plethora of smart home system solutions for various application domains that influence our daily life and the way we live. One example of CPHS applications is the smart home automation for the implementation of energy efficient thermal comfort control (EETCC) [2], which operates and controls the home appliances, devices, sensors, and actuators in a timely manner to assist people to live on their own comfortable, convenient, relax, restful, and pleasant.

Smart homes provide ambient environmental conditions for the residents to live, that means that the resident is the most important human factor. In other words, the human has a strong interaction with all the smart home systems and their surrounding environments. Unlike the previous works on smart homes and human interaction, Ooi et al. [3] present an adaptive model predictive control (MPC) based controller that is integrated into the existing EETCC system for the CPHS environment. One of the significant this work is that the adaptive MPC based controller can monitor the temperature in a real-time manner by using the sensed raw environmental data from the experimental house, iHouse. Meanwhile, Chen et al. [4] propose a human-centric smart home energy (SHE) management system that integrates ubiquitous sensing data from the physical and cyber spaces to discover the patterns of power usage and cognitively understand the behaviors of human beings. The relationship between them is established to infer users’ demands for electricity dynamically, and then the optimal scheduling of the home energy system is triggered to respond to both the users’ requirements and electricity rates.

Thermal comfort is an assessment of one’s satisfaction with the environment surroundings in which an individual is depending on factors such as indoor temperature, activity level, clothing, and relative humidity. Thermal comfort is an important goal of heating, ventilation, and air conditioning (HVAC) system design engineers. Today, the predicted mean vote/predicted percentage of dissatisfied (PMV/PPD) model [5] is well-established evaluating the environmental thermal comfort model and it is also assessed by the subjective evaluation (ANSI/ASHRAE Standard 55 [6]). The said environmental thermal comfort model is widely used in the calculation and evaluation of environments such as offices and classrooms for the groups of people. This leads to many people feeling either cold or hot in the built environment as it is supposed to be thermally comfortable for most people. However, in smart homes, the human thermal comfort takes an attribute for an individual resident as the unit of analysis rather than the groups of people. Compared to the environmental thermal comfort model, the human thermal comfort is a highly subjective feeling and hard to be measured objectively in a single person. Although PMV and ASHRAE Standard 55 are widely used as an indoor thermal comfort scale, they have not yet been able to fully elucidate the relationship between individual’s feelings and environment parameters, especially in smart home domains. Rupp et al. [7] review and propose the human thermal comfort in the resident building by considering light, noise, vibration, temperature, relative humidity, etc. However, it is not considered the human thermal comfort relationship in between the environmental parameters and the human physiological parameters. The relationship between the human thermal comfort and the physiological parameters should be investigated to get a better understanding of the mechanisms underlying the thermal comfort for automation control in smart homes.

For many decades, the human thermal comfort has been studied in terms of environmental factors and physiology parameters. Several sophisticated theories and objective indicators have been developed, such as the operative temperature, sufficient temperature, and effective standardized temperature. Luo et al. [8] explore the notion of comfort expectations and ask the question of whether the change as a result of long term exposure to mild indoor climates. Okamoto et al. [9] reveal new physiological markers of the response to indoor airflow sensation that airflow alter the feelings of the participants. Heart rate variability (HRV) as a predictive biomarker of thermal comfort. The result of this study suggests that it could be possible to design automatic real-time thermal comfort controllers based on people’s HRV. Both research teams from Nkurikiyeyezu et al. [10] and Zhu et al. [11] have been studied on the Electrocardiogram (ECG) data. The frequency-domain method is adopted to obtain the HRV results and to explore the human thermal comfort under different environments. The results are shown that the observation of different low frequency/high frequency (LF/HF) values under different situations, the air temperature has the most significant effects on the LF/HF values. These changes in the air temperature could easily lead to the excitation of the sympathetic nerve that could also promote the activities of the thermoregulatory effectors, i.e., thermal discomfort. Additionally, the relationships between the LF/HF, the thermal sensation and the thermal comfort are also revealed. Hasan et al. [12] propose the sensitivity of the PMV thermal comfort model with relative to its environmental factors and personal parameters using the wearable devices. It is found that the expected error range of PMV is high when the other parameters are ignored, such as clothing and metabolic rate. Besides that, Zhu et al. [13] focus on the dynamic thermal environment that gives an effect to the human thermal comfort. In [14], Salamone et al. describes a workflow for the assessment of the thermal conditions of users through the analysis of their specific psychophysical conditions overcoming the limitation of the physic-based model in order to investigate and consider other possible relations between the subjective and objective variables.

Over the previous researches, the human thermal comfort in the field of smart homes has reported the relationship between air temperature, air speed, and relative humidity with dynamic control of PMV and PPD value. However, the personal physiological is not obtained in a real-time manner and its correlation with environmental factors is not well-investigated, especially in smart home environment. Although many theoretical models that based on the PMV as an index of thermal comfort are the most commonly used and well-accepted in worldwide researchers, several studies pose that these models are not accurate for predicting the thermal sensation of residents in the buildings with natural ventilation, and these models tend to be underestimated or overestimated the actual conditions of thermal comfort.

With the continuous development of the Internet of things (IoT) paradigm, we can easily obtain human physiological data from the IoT devices, such as smart wearable devices, thermal cameras, medication equipment, and so on. Heart rate is one of the human physiological data that can be measured by using the smart wearable device daily and timely. The sensing technologies of the smart wearable device with the measured data are providing a novel opportunity to understand human behavior and intention. Besides that, the measured data is identical and unique to an individual person. However, the measured data forms a challenge to the relationship with the aforementioned environmental thermal comfort model, which cannot well-match to the personalized data. Many recent pieces of research are focusing on investigating the beneficiary of wearable technologies. For example, Kobiela et al. [15] aim the person’s individual momentary thermal sensation and comfort that involve physiological data, especially skin temperatures and Heart rate (HR)/HRV features based on the heart activity, then investigate those features to improve the prediction accuracy, which includes physiological data based on two data sources a smartwatch and a portable chest belt device. Georgiou et al. [16] investigate the smart wearable devices that *provide* the reliable and high-precision measurement compared to the classic heart rate measurement. Seshadri et al. [17] provide a comprehensive review of the applications of wearable technology for assessing the biomechanical and physiological parameters of the athletes. Wang et al. [18] put forward the human centric interactive clothing concept applied in daily wearable under the CPS framework.

In this paper, the objective is to introduce the CPHCS to take the advantage of human thermal comfort to improve the residents thermal comfort level while optimizing the energy consumption at the same time. The concept of CPS is adopted to the CPHCS for monitoring, controlling and maintaining the desired thermal comfort level in a timely manner with three actuators; air-conditioner, window, and curtain. The proposed CPHCS is also an extension of the EETCC with the human thermal comfort model, which can be measured by using the smart wearable device. Our contributions in this paper are:A significant part of this research is to use the commercial smart wearable devices as the measurement device incorporated with the other sensors and actuators to build and propose the generic CPHCS framework;Besides the environmental factors, the physiology parameter from the heart rate is well-studied and its correlation with the environmental factors, i.e., PMV, air speed, temperature, and humidity are deeply investigated to reveal the thermal comfort level of the plain air-conditioner (Air-con) and EETCC systems in the smart home environment; andThrough the questionnaire method, the subjective comfort level (SCL) of the human thermal comfort is directly obtained and verified with the thermal comfort level of the EETCC systems. In this way, a generic human thermal comfort model that can be applied to the CPHCS framework is attained in which the coefficients of this model can be fine-tuned to well-fix to the individual thermal comfort.

The organization of this paper is as follows. In Section 2, the background about smart homes, cyber–physical systems, EETCC, and human thermal comfort will be discussed. In Section 3, the proposed CPHCS framework is elaborated. Section 4 describes experiment setup and procedure. Besides that, the participant information, iHouse environment, experiment strategy and subjective comfort level. In Section 5, experiment results and discussion are provided. Section 6 summarizes the paper and provide some conclusions.

## 2. Background

### 2.1. Smart Homes

Smart Homes are home environments that incorporate ambient intelligence and automatic control that reacts to the behavior of its residents with various home appliances and devices. Smart Homes are one of the CPS application domains. The smart home is one of the key technologies to solve the problem of an aging society. In the future, a smart home will integrate into daily life with dedicated artificial intelligence, computational power, communication skills, monitoring, and controlling abilities needed to improve everyday activities. The interaction between people and home appliances will be devoted to improving comfort, health care, safety, security, and energy savings [19,20].

There are numerous researchers focus on a smart home in different domains. In [21] old aging people application is proposed in smart home. In [22], smart home is an application of ubiquitous computing in which the home environment is monitored by ambient intelligence to provide context-aware services and facilitate remote home control. The [23] paper introduces the concept of the smart home investigates technologies for smart homes, in which advanced technological systems that allow the automation of domestic tasks are developing rapidly. In [24], the authors propose a holistic framework that incorporates different components from IoT architectures introduced, to integrate smart home objects efficiently in a cloud-centric IoT based solution to contribute towards narrowing the gap between the existing state-of-the-art smart home applications and the prospect of their integration into IoT enabled the environment.

There is the current implementation of the smart home environments, one is Aware Home project [25], built-in 1999 from Georgia Tech. It is a living laboratory for research in ubiquitous computing for everyday activities. Another one is the experiment environment, iHouse that is used in this paper is located at Nomi City, Ishikawa Prefecture, Japan. It is a conventional two-floor Japanese style house featuring more than 300 sensors, home appliances, and electronic house devices that are connected using ECHONET Lite version 1.1 and ECHONET version 3.6.2. An ECHONET Lite system incorporates groups of devices with the same management of properties, security, and so on. Therefore, the largest area that ECHONET Lite can manage is referred to as a domain. A domain will be specified as the range of controlled resources (home equipment, appliances and consumer electronics, sensors, controllers, remote controls, etc.) present within the network range determined by ECHONET Lite. A system is defined as that which performs communication and linked operations between devices and the controllers that monitor/control/operate them and between devices themselves. A system lies within one domain and does not extend over a number of domains. A domain includes one or more systems. Thus, the same device or controller can exist in more than one system. When connecting a system to another system lying outside the domain, an ECHONET Lite gateway is used as an interface. iHouse is developed for the research of smart home environment monitoring, energy-saving, and human comfort. iHouse is named because of standing in Ishikawa, internetted, inspiring, and intelligent house, which is an advanced experimental environment for future smart homes in Japan, and it has been implemented according to Standard House Design by Architectural Institute of Japan.

### 2.2. Cyber–Physical Systems

Cyber–physical systems (CPSs) are defined as tight integration of computation, communication, and control with deep interaction between physical and cyber elements in which embedded devices, such as different sensors and actuators, are wireless or wired networked to sense, monitor and control the physical world [26,27]. J.F. He [28] interpret CPS as controllable, credible, and scalable networked physical equipment systems, which is in-depth integration of computation, communications, and control ability based on environmental perception. CPS enables the cyber world to interact with the physical world to monitor and control the intended parameter on a real-time basis. The systems of CPS represent the intersection of several system trends, such as real-time embedded systems, distributed systems, control systems, and networked wireless systems. Liu et al. [29] review that the most critical part is the physical system and the core part is the cyber system. In recent years, CPS has enlivened many fields of manufacturing, automotive systems, military systems, smart homes, smart transportation systems, power generation and distribution, energy conservation, HVAC system, aircraft, and smart city. In a typical CPS application, sensor nodes collect information from the physical world as a source of CPS input. Upon receiving the input information, a controller makes a corresponding decision by computing, and actuators perform a relevant action in the physical world through the closed-loop feedback.

CPS is the system of systems where its physical and computational resources are strictly interlinked together. In some home domains, cyber–physical home system (CPHS) offers residents to live more comfortably, conveniently, cost-effectively, and more securely using the CPS approach. A typical CPHS, where it is comprised of the cyber world, physical world, and the communication network in between them. The control domain, which includes data logging and supervisor controller is part of the cyber world while the sensor domain and actuator domain are part of the physical world. Both cyber and physical worlds can be linked together by networks and communication protocols that are not limited to wired networks but also wireless networks. One example of CPHS applications is smart home automation for the implementation of EETCC system [2], in which the home appliances, devices, sensors, and actuators are synergized in a timely manner to assist people to live on their own comfortable, convenient, relax, restful, and pleasant.

### 2.3. Human Thermal Comfort

Thermal comfort is described as the state of the mind that expresses satisfaction with its thermal surrounding. Assessing thermal comfort is primarily regulated using models based on static heat model transfer equations. P.O. Fanger has been proposed the predicted mean vote/predicted percentage of dissatisfied (PMV/PPD) model in 1970 [5]. This model has been presented by ISO-7730 (2005) [30]. The static thermal models, however, have some limitations. For example, the PMV/PPD model is based on laboratory experiments on adults in highly controlled thermal chambers for a relatively extended period. It is not suitable for different age range of people at home. Halawa et al. [31] review the studies on adaptive thermal comfort and look critically at the foundation and underlying assumptions of the adaptive model approach and its findings. Craenendonck [32] review the experiments of human thermal comfort in controlled and semi-controlled environments.

Although the PMV/PPD model gives us a way in judging the thermal comfort level, the human’s subjective evaluation is essential. In this paper, we use the ASHRAE 55 [6] to make the subjective evaluation level into seven-level evaluation, as shown in Table 1. It contains seven thermal sensation levels: “cold (−3)”, “cool (−2)”, “slightly cool (−1)”, “neutral (0)”, “warm (1)”, “slightly warm (2)” and “hot(3)”, respectively. In [33,34,35], the subjective comfort evaluation methods are given in smart homes. This subjective evaluation level is modified as a subjective comfort level (SCL) to be used for the participant to answer their direct thermal sensation via the online questionnaire with the intention to study the difference in between human’s subjective thermal comfort and system’s thermal comfort level.

According to the standard ISO 7730 of the indoor environment comfort index, in summer season the environment temperature is in range of 23 and 28 °C, relative humidity is in range of 30% and 70%, body vertical temperature difference is less than or equal to 3 °C, average wind speed is less than 0.25 m/s. In winter season, the environment temperature is in range of 20 and 24 °C, relative humidity is in range if 30% and 70%. During these specification ranges, a human will feel comfort in indoor environments.

On the other hand, the predicted mean vote (PMV) is a particular combination of air temperature, mean radiant temperature, relative humidity, air speed, metabolic rate, and clothing insulation, that is formally shown in Equation (Equation 1).
(1)$$PMV=f_{pmv}\left(t_{a},\bar{t_{r}},v_{ar},p_{a},M,I_{cl}\right)$$
where $t_{a}$ is the air temperature, $\bar{t_{r}}$ is the mean background radiant temperature, $v_{ar}$ is the air velocity, $p_{a}$ is the humidity level, *M* is the metabolic rate, and $I_{cl}$ is the human’s clothing insulation factor.

### 2.4. Energy Efficient and Thermal Comfort Control

The energy efficient thermal comfort control (EETCC) algorithm is a supervisory rule-based control controller developed for smart home [36]. The EETCC algorithm is a thermal comfort controller that utilizes the actuator that uses the least energy consumption to maintain thermal comfort in the room. For example, when the outdoor air temperature is lower than the indoor temperature while the intended action is to cool the room, the EETCC algorithm will open the window to allow the cold outdoor air to lower the temperature in the room instead of utilizing the air-conditioner. Besides, the EETCC algorithm uses the states of actuators as different PMV categories to determine which combination of actuators to turn on or off. The PMV, PPD, and DR at every iteration to determine the state of actuators that satisfy the target thermal comfort demand while consuming the least energy. Furthermore, there is a timer in the EETCC algorithm to prevent frequent actuation to reduce the tear and wear of the actuators in the room.

As the experimental smart home, iHouse contain various types of networked sensors and actuators that provides the required feedback parameters and output controls to the proposed temperature controller. In order for the temperature controller to be able to communicate with the networked sensors and actuators in the iHouse, an ECHONET Lite capable system has to be developed to translate the necessary control signals and feedback sensor data to the appropriate formats. The EETCC system mainly provides ECHONET Lite protocol translation, data processing while supporting real time device management and data logging. There are two revisions of the EETCC system that is developed, where the one version is written in C language and the other one version is rewriten in Python language. For this study, the C language version is chosen. Two PMV categories are considered in the EETCC control algorithm, which one is category A: −1<PMV<1 and another one is category B: −0.5≤PMV≤0.5. Then we considered three different actuators that can be used to control the thermal comfort in the room, which are the air-conditioner, window, and curtain. These three actuators can be categorized into eight different actuation profiles, where each profile is a combination of the actuation state of the actuators in the room. However, only six combinations are implemented. Two of the states are removed from the eight possible combinations as both of the states involves turning on the air-onditioner and opening the window at the same time. These states are not logical as opening the window while cooling or heating the room with an air-conditioner is inefficient as the heat exchanges due to convection between the indoor and outdoor environment would occur and reduce the capability of the air-conditioner to cool or heat the room. Hence, increasing the time taken to cool or heat the room to a certain temperature at the same time will increase the energy usage by the air-conditioner. The remaining six states are shown in Table 2. The result of the control state and human thermal comfort will be discussed in Section 5.4.

## 3. Cyber–Physical Human Centric Systems

Human and system cannot be separated. A system is designed for human requirements; at the same time, a human using a system to make his life better. With the high development of CPS and IoT technology, the performance inner requirements of human centric keep on increasing. Up to now, previous studies have shown that interaction is essential between the CPS and humans. A cyber–physical social system (CPSS) in Liu et al. [37] regards human factors as a part of a system instead of placing them outside the system boundary. Higashino and Uchiyama [38] proposed the human centric cyber–physical system application where the effects of human activities are taken into consideration for designing and developing CPS based societal systems. Schirner et al. [39] proposed a prototyping platform and a design framework for rapid exploration of a novel human-in-the-loop application serves as an accelerator for new research into a broad class of systems that augment human interaction with the physical world. Sowe et al. [40] presents people in loop of cyber–physical–human systems.

Summarizing the current about human centric CPS research, there are two problems. One is the device, system, or platform can provide efficient and effective performance based on the desired value, but human need and social do not consider at all. Another is many current research works consider human factors into the system design, but no tight synchronization between human preference and the system. To meet these demands, cyber–physical human centric systems (CPHCSs) can offer numerous opportunities. As Figure 1 show, CPHCS is mainly interactions between human and CPS. Human has lots of factors, such as preference, physical health, need and want, social norm, role and knowledge, personality trait, background, and so on. CPHCS is consist of interconnected systems (cyber world, physical world, and human) interaction to each other across space and time, and allowing other systems, devices, and data streams to connect and disconnect.

## 4. Experiment Study

In this section, we present the experiments to study the correlation between environmental parameters and individual heart rate under the different subjective comfort level with EETCC uninterruptedly collected the environmental parameters and system comfort during the experiment to calculate the calculation results. At the same time, participants wear wearable devices to collect heart rate data continuously. Participants also completed the completion of the subjective questionnaire during the experiment.

### 4.1. Experiment Setup

#### 4.1.1. Content and Participants

Consent is obtained from all participants before the subjective questionnaire and the measurements. All participants agree to participate in the survey and experimental processing. It should be noted that this study mainly focused on the usage of personal thermal comfort in a real home scenario, that is involved in the use of smart watch to observe the heart rate only. All involved systems are widely used or studied in the real world, and they do not cause any harm to people.

There were six participants (two female adults, three male adults and one female child ), with the adult participants average age, height, weight and BMI of 26.8, 171.0 (±2) cm, 64.4 (±0.5) kg, 21.72 (±0.2) kg/m^2^. Detailed physical information about the research participants is shown in Table 3. All participants had no physical defects, lack of sleep, depression, and other conditions.

#### 4.1.2. iHouse Environment

We chose iHouse the second-floor Bedroom 1 as this experiment environment. Bedroom 1 is 5.0 m length, 4.1 m width, 2.4 m height. As shown in Figure 2, there are two windows, and one curtain in Bedroom 1 which could be controlled. The experiment’s implementation is from May to July 2019 in Nomi City, Japan; during this time the average normal highest outdoor temperature was 24.7 °C, the average normal lowest outdoor temperature was 17.7 °C. Main sensors and wearable devices are shown in Table 4. To collect human physiological data, the authors compared several portable wearable and health monitoring devices available in the market, which were popular and common due to powerful functionality, affordable prices, and lightweight features. Considering our requirements on the specification of individual heart rate data, the Apple Watch Series 4 was adopted.

#### 4.1.3. Subjective Comfort Level

In the subjective experience record section, we used random submission and passive submission hybrid mode. The random submission mode is that the participant could submit records online when they feel thermal comfort changing. The passive submission mode is a record provided when the system’s physical environment changes, for example, the air-conditioner is turned off, the windows are opened, and so on. The Google Forms open-source application is used to share the web link of the questionnaire. Each participant is independently filled in at different terminals (personal computers or personal smartphones), ensuring the accuracy of the experimental results, and removing interference. The questionnaire record includes: (1) date and time; (2) number of participation; (3) thermal sensation and comfort; Scales of subjective comfort data records example are presented in Table 5. During the experimental test period, 225 individual comfort data records are collected. Summary of the subjective comfort data, which top three are 38.2% neutral, 20.4% slightly cool and 14.7% slightly warm.

### 4.2. Experimental Procedure

The experiment was divided into two sets. The first set was two participants who manually adjusted the air-conditioner setting every 30 minutes, automatically recorded the heart rate with a wearable device, and filled in SCL card online during the operation. The second set is based on EETCC’s automatic control of the comfort level of the environment. The wearable device automatically recorded the heart rate, and the subjective comfort was filled in online when the subjective comfort changed. Each collection was divided into two sections, morning session (10:00 AM–11:30 AM) and afternoon session (14:00 PM–16:00 PM), shown in Table 6. After the end of each session, we turned off the air-conditioner, closed the doors and windows, and let the indoor environment without the system adjustments and human interference.

Six subjects participated in experiments from May to July 2019. Morning and afternoon sessions lasting two hours each were scheduled. It should be noted that participants were volunteers and were aware of the purpose and procedure of data collection. The process defined for the tests required the users to carry the wearable devices under normal house conditions, and no particular activity was determined for the experimental. Each participant was asked to arrive at iHouse, the experimental building, 30 min before starting the experiment. In the first 15 min, the participant took the wearable device for adapting to the comfort and usability of wearable devices. At that same time, we told the participant about the experimental considerations and experimental processing and requirements. During the next 15 minutes, the participant was guided in the experimental room to have a rest with sitting or reading before starting the experiment and recording the data. Furthermore, the participants were conducting their usual home activities during the tests. These activities were comprised of reading or writing, working on a computer, etc. The indoor and outdoor air temperature, indoor and outdoor relative humidity, indoor and outdoor air speed were measured continuously by EETCC every 10 s.

## 5. Results and Discussion

In this section, we divided the experimental results into four parts. The first part is the statistical results of the participants’ heart rate collected by the wearable device. The correlation coefficient between the environmental parameters and the participants’ heart rate is mentioned in the second part. The third part is about the thermal comfort evaluation of the system compared with the subjective. The fourth part has suggested the relationship between comfort parameters and the control state of the smart home system.

### 5.1. Statistical Results

The statistical result of the heart rate with the mode value is 70, the median value is 74, the average value is 75.78, the variance is 104.95, and the standard deviation σ is 10.25. Comparison based on the distribution of different heartbeat data for males and females is shown in Figure 3. There are 68.83 % heart rate data records concentrated in the interval of 60–70 and 71–80.

### 5.2. Correlation Co-efficient among Environment Parameters and Heart Rate

The correlation between heart rate and environmental factors is tested using Pearson Correlation (*r*), which is shown in Formula (Equation 2). The results of different subjective comfort levels are shown in Figure 4a morning session and Figure 4b afternoon session.
(2)$$r=\frac{\sum_{i=1}^{n}\left(x_{i}-\bar{x}\right)\left(y_{i}-\bar{y}\right)}{\sqrt{\sum_{i=1}^{n}{\left(x_{i}-\bar{x}\right)}^{2}{\left(y_{i}-\bar{y}\right)}^{2}}}$$
where *x* is the heart rate and *y* is environment parameters include temperature indoor, temperature outdoor, relative humidity indoor, relative humidity outdoor, air speed indoor, air speed outdoor, PMV, and PPD.

Results are quite revealing in several ways. The morning session is shown in Figure 4a, the temperature indoor is most relevant to the heart rate at the warm SCL, and the correlation co-efficient r=0.64. The indoor relative humidity is the most pertinent to the heart rate at the slightly warm SCL, r=0.36. The air speed indoor is the most relevant to the heart rate correlation value r=0.30 at warm the SCL. There is a significant positive correlation between heart rate and indoor parameters under warm SCL. The correlation between heart rate and outdoor environmental parameters is positively correlated with the outdoor temperature at the cold comfort level of r=0.14. Outdoor relative humidity at cold SCL is r=0.27. Outdoor air speed is r=0.30 at warm SCL. Meanwhile, the negative correlation coefficient of outdoor relative humidity is more prominent in the warm SCL; the value of the outdoor relative humidity correlation coefficient is r=−0.46.

Turning now to the experimental results on the afternoon session, shown in Figure 4b. The indoor environmental parameters were more prominent under different subjective comfort levels. The correlation coefficient between indoor temperature and heart rate is r=0.36 at hot SCL. The indoor relative humidity and heart rate correlation coefficient is r=0.27 at cool SCL. Indoor air speed correlation with the heart rate is r=0.25 at cold SCL. On the other hand, the outdoor parameters correlation of temperature is r=0.41 at cold SCL, relative humidity was r=0.48 at warm SCL, outdoor air speed is r=0.10 at warm SCL, and r=−0.49 at hot SCL.

### 5.3. Thermal Comfort Assessment

Referring to Berkeley’s research results in [41], we define proximate comfort zone, cool zone, and warm zone in this paper’s results shown figures. In experiment Set 1, the setting of the air-conditioner is controlled by the participants. Participants set the air-conditioner temperature to 20 or 25 °C, and changed every 30 min. During the time, participants recorded the subjective comfort information.The results that PMV represents air speed against operative temperature without EETCC are shown in Figure 5a,b. In the morning session, PMV is 58.8% in the comfort zone, 41.2% in the cool zone, and 0% in the warm zone. In the afternoon session, the PMV is 52.5% in the comfort zone, 45.8% in the cool zone, and 1.7% in the warm zone.

In experiment Set 2, indoor temperature and air speed data were automatically recorded by EETCC. Based on Berkeley’s calculation comfort zone, we set the clothing insulation as indoor summer clothes, metabolic rate is reading while sitting, and the average relative humidity is 50%, and then draw the comfort zone as shown in Figure 6a,b. The comfort zone is highlighted in green color, where its PMV value is between −0.5 and 0.5. The warm zone and cool zone are emphasized in blue color and red color. The draft zone is shown up in yellow color.

The morning session result is shown in Figure 6a. There is 64% PMV value in comfort zone, 34.5% in the warm zone, and 1.5% in cool zone. In the afternoon shown in Figure 6b, the PMV value in the comfort zone is 67.8%, 32% in the warm zone, and 0.2% in cool zone. There is no PMV value in the draft zone, whether in the morning or afternoon.

From Figure 7, we can see that relative frequency in different SCL scale. In the neutral range, there is 41.4% in the morning and 66.6% in the afternoon. Average the slightly cool, cool, and cold SCL scale in the cool zone, there are 12.3% in the morning and 11.7% in the afternoon. For the warm zone with average, the slightly warm, warm, and hot SCL value, which are 46.3% in the morning and 21.8% in the afternoon.

### 5.4. Control State and Thermal Comfort

To consider the difference between thermal comfort and thermal sensations in different indoor temperatures scale, the average thermal comfort data is shown in Figure 8a,b. The gray zone indicates the PMV values between −1 and 1. The yellow zone indicates the PMV values between −0.5 and 0.5. In Figure 8a, the morning session, the average thermal comfort is greater than 1 when the indoor temperature is greater than 28.5 °C and the indoor temperature in the range from 27.5 to 28.4 °C. Each scale of the thermal sensation is unevenly distributed. The indoor temperature scale from 25.5 to 26.4 °C and scale from 26.5 to 27.4 °C, those two scales are in thermal comfort yellow zone in thermal sensation scales in Slightly Cool, Neutral, Slightly Warm and Warm. In Figure 8b the afternoon session, the indoor temperature scale from 25.5 to 26.4 °C in thermal comfort yellow zone with the thermal sensation from Cold to Hot.

According to the definition of the previous Section 2.4, total of control states is six. In Figure 9a,b, we show the relationship between the control state and PMV and SCL with time continues. The gray zone indicates the PMV values between −1 and 1. The yellow zone indicates the PMV values between −0.5 and 0.5. The blue line is the highest frequency control state. The black line is the highest frequency PMV value. The red line is the highest frequency SCL value from the six participators.

### 5.5. Discussion

Our studies concluded that the correlation between single heart rate and environmental parameters in the smart home. The results show that the human heart rate has more association with indoor temperature, indoor relative humidity, and indoor air speed in the warm zone (including slightly warm, warm) in the morning. The reason for this phenomenon is that the air-conditioner is turned on in the morning to adjust the temperature. The indoor temperature at this time has a specific outdoor temperature and no ventilation. This phenomenon changed in the afternoon, and the heart rate in the afternoon was shown more correlation to the indoor temperature, indoor relative humidity, and air speed in the cold zone. This is because EETCC has achieved proper comfort in the room after adjustment in the morning. When entering the experimental scene in the afternoon, the starting value of comfort is closer to subjective comfort. Our results suggest a possibility that is achieving system thermal comfort in a short period or maintaining system thermal comfort over a long period has a crucial role in producing human thermal comfort adjustment of the heart rate reference system.

Furthermore, the use of EETCC can significantly improve the human thermal comfort level. For example, in Section 5.3, the comfort zone ratio increased by an average of 10.25% under the control of EETCC in the morning session and the afternoon session. However, the presence of the warm zone increased by an average of 32.4%, and the present of cool zone decreased by 42.65%. This is because the first principle of EETCC is energy efficiency. The data collected from the environmental sensors is calculated, and if it is within the system’s comfort zone, the existing controller state is maintained unchanged.

From the control state and thermal comfort result in Section 5.4, although the EETCC well performs the control strategy of system comfort, the human thermal comfort still is not considered. From the analysis of the result shown in Figure 9b, the average thermal comfort is in a stable state, the state change can be avoided, and the previous state can achieve the double demand of the human thermal comfort and the energy efficient to the maximum extent.

## 6. Conclusions and Future Works

Based on the studies on human thermal comfort in this paper, several conclusions can be drawn. Creating the environmental thermal comfort model for the indoor environments should not be the ultimate goal for the thermal comfort services in the smart homes. Human thermal comfort that comprises of the subjective thermal level of human and thermal comfort level of system is more suitable for the individual human. Indeed, the thermal comfort control systems are beneficially operating to well-fit to the human’s comfort preference with the guaranteed indoor air quality for healthcare smart home environments. This paper has also introduced the CPHCS framework, which consists of the generic of human thermal comfort model and EETCC system.

Experiment results reveal that there is a tight correlation between the environmental factors and the physiology parameter (i.e., heart rate) in human thermal comfort. Through this experiment, this paper also can conclude that the current EETCC system is unable to provide the precise need of thermal comfort to the human’s preference. However, the experiment results discover that the relationship between the human thermal comfort and the physiological parameters that we can obtain data from conventional wearable devices has a tied correlation, and this gives a better understanding of a novel solution underlying the thermal comfort for automation control in smart homes.

Many unanswered questions remain in this paper should be addressed by the future studies. For example, what is the appropriate solution for the CPHCS, which uses the human thermal comfort with human physiological data from wearable devices. How the human thermal comfort model is updated and specified for an individual human with different range of age in smart homes? Does the questionnaire for the human thermal comfort environments should be revised? In conclusion, understanding human response in the aspect of thermal comfort and designing the realization of CPHCS framework is an ultimate goal of this research, but still there is a long way to go.

## Figures and Tables

**Figure 1 sensors-20-00372-f001:**
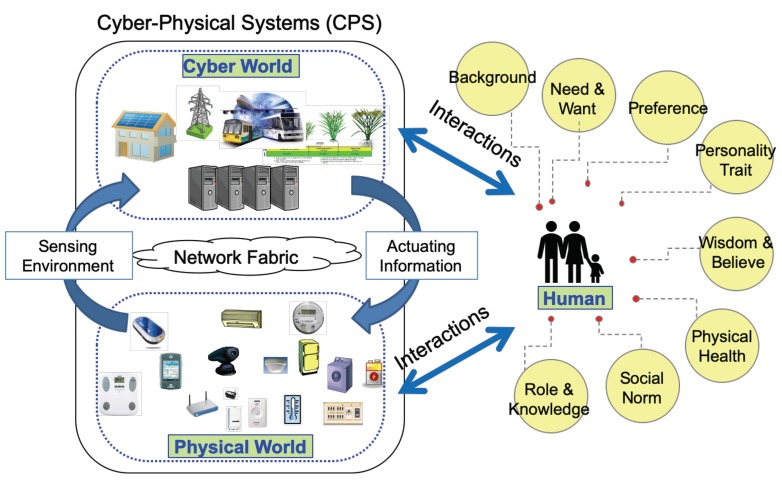
The schematic diagram of cyber–physical human centric system (CPHCS).

**Figure 2 sensors-20-00372-f002:**
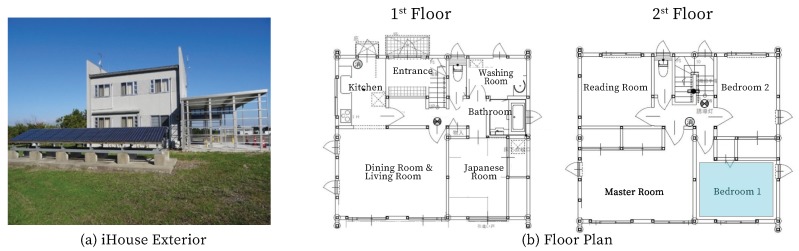
iHouse exterior and architectural plan.

**Figure 3 sensors-20-00372-f003:**
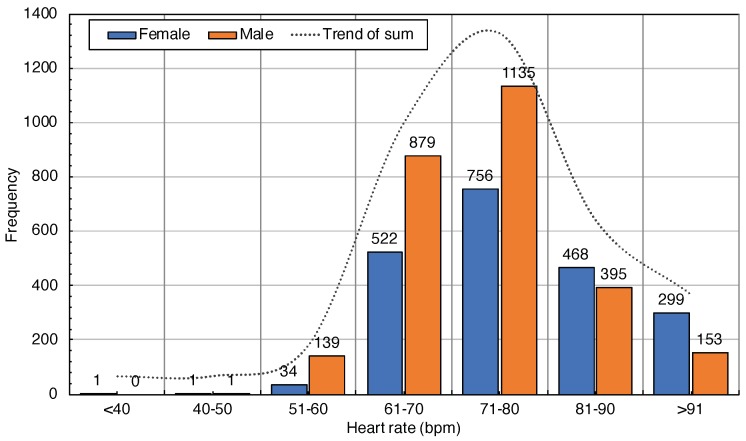
Distribution of heart rate.

**Figure 4 sensors-20-00372-f004:**
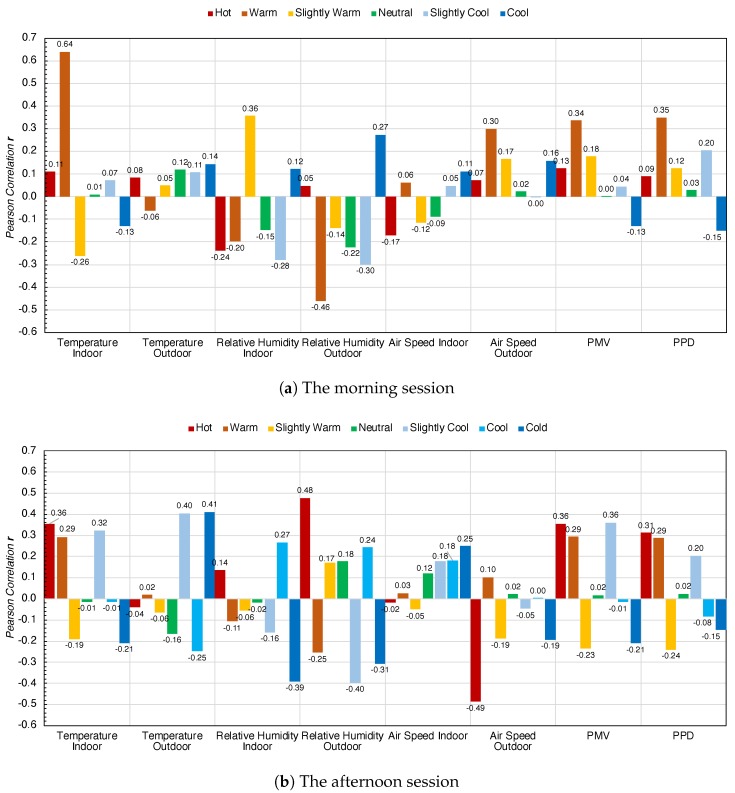
Average Pearson correlation *r* between the environmental parameters and heart rate.

**Figure 5 sensors-20-00372-f005:**
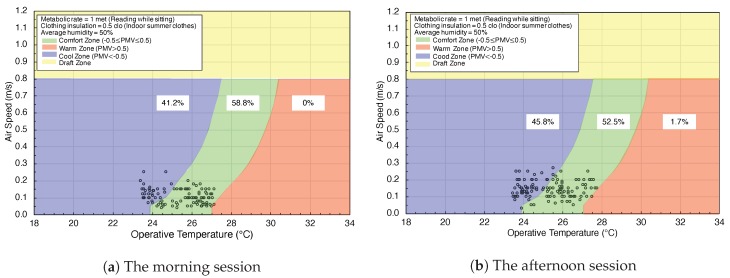
PMV represents air speed against operative temperature without EETCC.

**Figure 6 sensors-20-00372-f006:**
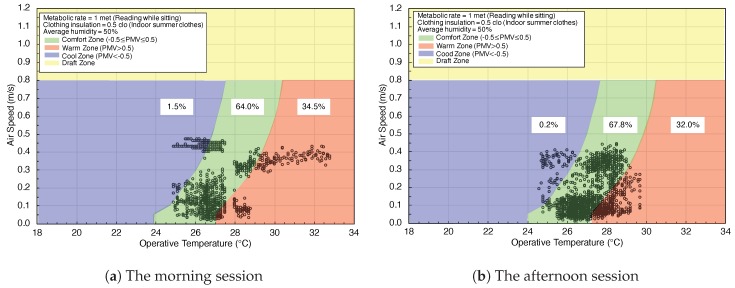
PMV represents air speed against operative temperature with EETCC.

**Figure 7 sensors-20-00372-f007:**
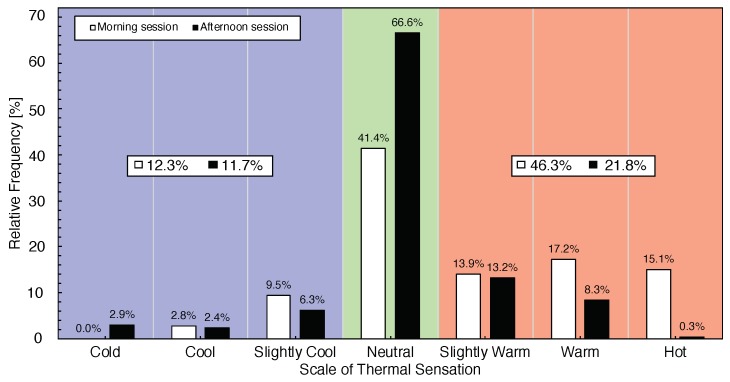
Subjective Comfort Level represents air speed against operative temperature.

**Figure 8 sensors-20-00372-f008:**
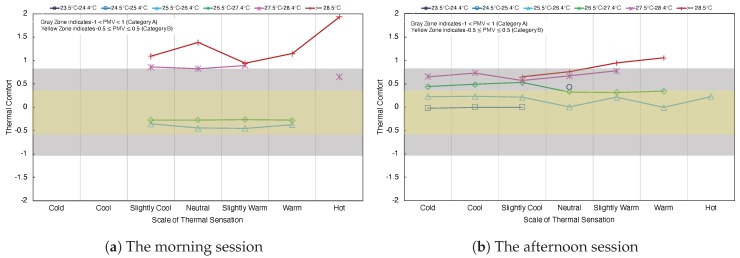
Thermal sensation and thermal comfort at different indoor air temperatures.

**Figure 9 sensors-20-00372-f009:**
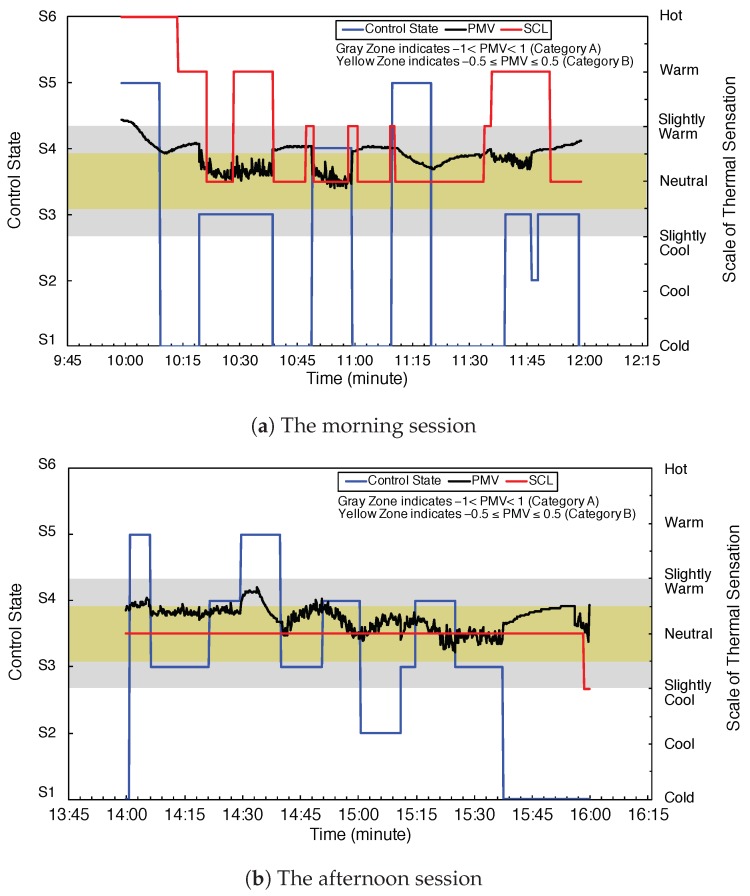
Control state, PMV, and subjective comfort level.

**Table 1 sensors-20-00372-t001:** The average human’s comfort degree on the 7-point ASHRAE scale.

Cold	Cool	Slightly Cool	Neutral	Slightly Warm	Warm	Hot
−3	−2	−1	0	+1	+2	+3

**Table 2 sensors-20-00372-t002:** States of the actuators.

State	Air-Conditioner	Window	Curtain
S1	0	0	0
S2	0	0	1
S3	0	1	0
S4	0	1	1
S5	1	0	0
S6	1	0	1

0: Off/Close; 1: On/Open.

**Table 3 sensors-20-00372-t003:** Brief information on participants.

NO.	Gender	Age (y)	Height (cm)	Weight (kg)	BMI (kg/m^2^)	Test Period	Avg. Indoor Temperature (°C)	Avg. Relative Humidity (%)
F1	Female	38	174	59	19.5	May 30 May 31	25.2 26.5	48.5 45.9
F2	Female	23	154	47	19.8	July 19	25.9	49.6
M1	Male	26	170	56	19.4	May 30 May 31	25.2 26.5	48.5 45.9
M2	Male	23	175	65	21.2	June 28	27.1	50.9
M3	Male	24	182	95	28.7	June 28	27.1	50.9
C1	Female	8	137	27	14.4	July 21	28.3	51.0

**Table 4 sensors-20-00372-t004:** Brief information on main sensors and wearable device.

Type	Name	Range	Parameter
Indoor temperature sensor	SHT75 digital sensor	[−40, 125] °C ± 0.3 °C	14-bit ADC signal processing
Relative humidity sensor	SHT75 digital sensor	[0,100]% ± 1.8%	14-bit ADC signal processing
Air velocity sensor	hot-wire anemometer sensor	[0.015,5] m/s ± 0.2%	-
Wearable device	Apple watch series 4	[30, 210] *bpm*	64-bit dual-core CPU processor, 16 GB capacity

*bpm*: beats per minute.

**Table 5 sensors-20-00372-t005:** Subjective comfort data record structure example.

Participation Number	Date and Time	Thermal Sensation	Scale
F1	2019/05/30 10:27:01	Hot	3
M1	2019/06/28 14:05:34	Warm	2

**Table 6 sensors-20-00372-t006:** Experiment Sets

Set	Session	Participant	Contents	Total of Datasets	Total of Samples
Set1	Morning Afternoon	F1, M1	Air-con is set to 20 and 25 °C alternatively in every 30 min Fill the SCL card in every 30 min	4	2880
Set2	Morning Afternoon	F1, F2, M1 M2, M3, C1	Air-con is controlled by EETCC automatically Fill the SCL card in any time	12	8640

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
