# Peer review of "Study of Human Thermal Comfort for Cyber–Physical Human Centric System in Smart Homes"

_sensors, 2020, doi:10.3390/s20020372_

Round 1

Reviewer 1 Report

Although the domain of this experimental research it is very interesting, I have to submit to the attention of the Authors that the number and variety of data collected doesn't seem to be enough to reach a concrete goal, e.g. the range of age and the number of partecipant isn't enough to represent a real scenario. Also the idea that the heart rate measure by itself can drive the real time tuning of the HVAC system, does't seem to be applicable, cause of the huge numbers of different environmental variables affecting the real scenario. More in general, also the hypothesis that in a real application, the users should dress wearable sensors all the time, while into their own house doesn't look appealing in the real life. Based on the result exposed isn't clear enough how the data collected, even if affected by the before mentioned issues, show an effective correlation between heart rate and general weil being conditions. I suggest also to review and improve the english exposition.

Reviewer 2 Report

line 93 :"several studies pose that ..." provide references

provide information concerning the sensor communication protocols

correlation between heart rate and environmental parameter is quite low since very often is below 0.5

Reviewer 3 Report

This is a well-written, well-organized and well-illustrated paper.

The presentation of the technique and the characterization of the results obtained indicate that the method is quite suitable and could in fact be positively applied in the design of comfortable indoor environments based on the data obtained from the sensors.

I suggest to better describe the contribution of this paper compared to that of the same authors recently published.

Author Response

    We would like to appreciate you taking the time to review this manuscript significantly. We appreciate all your comments and suggestions! We tried our best to improve the manuscript and made some changes to the manuscript. These changes will not influence the content and framework of the paper. Several responses to the comments are given in detail below. And we marked in red in the revised manuscript. We appreciate the editor’s and reviewer’s warm work earnestly and hope that the correction will meet with approval.

Sincerely yours,

Fang Yuan

Reviewer 4 Report

The paper describes a Cyber-Phsysical Human Centric System (CPHCS) framework used to improve thermal comfort level and while optimizing the energy consumption. In the framework was considered also the heart rate as a biometric parameter:  the experimental results reveal that there is a tight correlation between the environmental factors and the heart rate in the aspect of human perception.

Please, revise the paper considering the following aspects:

I suggest to thoroughly review the introduction providing a series of references concerning the key issue. For example, one of the result of the paper is the tight correlation between heart rate, environmental variables and thermal human perception. In relation to this specific aspect, you have to improve your reference section by considering some reference studies. Among the other, I suggest the following papers that consider the importance of variables useful to define the Thermal Sensation of user: Kobiela, F., Shen, R., Schweiker, M., & Dürichen, R. (2019, September). Personal thermal perception models using skin temperatures and HR/HRV features: comparison of smartwatch and professional measurement devices. In Proceedings of the 23rd International Symposium on Wearable Computers (pp. 96-105). ACM. Salamone, F., Belussi, L., Currò, C., Danza, L., Ghellere, M., Guazzi, G., ... & Meroni, I. (2018). Application of IoT and Machine Learning techniques for the assessment of thermal comfort perception. Energy Procedia, 148, 798-805.

In line 30, please use the capital letters considering that is an acronym: “Efficient Thermal Comfort Control”

Line 36: same as above for “Model Predictive Control”

Lines 40, 48, 192, : same as above

Line 161: please add more details or a reference to the ECHONET communication protocol

Line 295: I wonder if the sentence “..while no chemical, biological, or medical test were involved” is correct considering that you acquired heart rate variable.

Table 3: please add the u.m. to age: “y”.

In line 314: please add more details about the frequency of data acquisition in passive mode.

In line 316: please add more details about how the form was shared

In line 328: have you considered an acclimatization period before the conduction of the test? If so, please, add more details.

In line 330: I wonder if “be natural” is a correct description of the environment without people.

In lines 356: why you have considered these variables? Please describe better

In line 375: please describe better experiment set 1

In line 437-438: this sentence is arguable, please provide a better description

In line 442: also the “strong” adjective used is arguable considering the small amount of available data.

Conclusion section need to be improved.

Round 2

Reviewer 4 Report

Considering the revised version of your paper, one major issue persist, which may render the whole approach in question in vain:

In relation to your sentence reported in line 352 and here below for your convenience, you have to consider that the acclimatization period it must be inside the experimental room and not outside it: "Before 30 minutes of every experiment, each participant will be asked to have a rest, like sitting and reading outside of the experimental room to obtain more stable and reliable heart rate."

Please verify/revise your sentence.

Two other minor issues are to be considered:

In table 3: please add "relative" to "humidity" The same is to be done in table 4: you have to consider "Relative humidity" instead of simply "humidity".

Round 3

Reviewer 4 Report

A minor issue persists: in table 4 you have to replace "wind speed sensor" with "air velocity sensor".